# p66shc siRNA-Encapsulated PLGA Nanoparticles Ameliorate Neuropathic Pain Following Spinal Nerve Ligation

**DOI:** 10.3390/polym12051014

**Published:** 2020-04-29

**Authors:** Nara Shin, Hyo Jung Shin, Yoonyoung Yi, Jaewon Beom, Wonhyung Lee, Choong-Hyun Lee, Dong Woon Kim

**Affiliations:** 1Department of Medical Science, Chungnam National University College of Medicine, Daejeon 35015, Korea; 2Department of Anatomy, Brain Research Institute, Chungnam National University College of Medicine, Daejeon 35015, Korea; 3Department of Pediatrics, College of Medicine, Hallym University and Gangdong Sacred Heart Hospital, Seoul 05355, Korea; 4Department of Rehabilitation Medicine, Seoul National University Bundang Hospital, Seongnam 13620, Korea; 5Department of Anesthesia and Pain Medicine, Chungnam National University Hospital, Daejeon 35015, Korea; 6Department of Pharmacy, College of Pharmacy, Dankook University, Cheonan 31116, Korea

**Keywords:** neuropathic pain, spinal nerve ligation, p66shc, PLGA nanoparticle, autophagy, proinflammatory mediators

## Abstract

p66shc, a member of the shc adaptor protein family, has been shown to participate in regulation of mitochondrial homeostasis, apoptosis, and autophagosome formation. The present study was performed to investigate whether p66shc siRNA-encapsulated poly(d,l-lactic-*co*-glycolic acid) nanoparticles (p66shc siRNA-PLGA NPs) can attenuate spinal nerve ligation (SNL)-induced neuropathic pain in rats. The SNL-induced pain behavior was decreased in the p66shc siRNA-PLGA NP-treated group compared with the scrambled siRNA-PLGA NP-treated group. In the L5 spinal cord of the p66shc siRNA-PLGA NP-treated group, expression levels of phosphorylated p66shc, cleaved caspase-3, p62, and PINK1, as well as microglial activation, were also decreased. In addition, p66shc knockdown using p66shc siRNA reduced the expression levels of cleaved caspase-3, p62, and PINK1, as well as proinflammatory mediators in the H_2_O_2_-treated HT22 neuronal cells. These results suggest that downregulation of p66shc expression in the spinal cord using p66shc siRNA-PLGA NPs could reduce the SNL-induced neuropathic pain by attenuating the SNL-induced aberrant autophagic, mitophagic, and neuroinflammatory processes in rats.

## 1. Introduction

Neuropathic pain, characterized by symptoms of allodynia and/or hyperalgesia, has been suggested to be a consequence of alterations in neuronal activities, immune responses, and neuroinflammatory processes [1,2]. Although many underlying mechanisms, related to the onset and development of neuropathic pain, have been suggested, mitochondrial dysfunction may be one of the most important pathophysiological mechanisms of neuropathic pain [3,4]. Our and other previous studies reported that mitochondrial reactive oxygen species (ROS) levels are increased in the neuropathic spinal cord, and that removal of ROS in neuropathic pain can lead to recovery of central sensitization and hyperalgesia to normal levels [5,6,7,8]. In addition to mitochondrial functions and ROS, autophagy has been regarded as one of the most important mechanisms for understanding neuropathic pain. Many studies suggest that imbalance of autophagy following peripheral nerve injury leads to changes in neuronal cell functions and neurodegeneration in the spinal cord as well as the development of neuropathic pain [9,10,11,12,13].

p66shc, a member of the shc adaptor protein family, plays roles in sensing redox stress [14]. When phosphorylated, p66shc is translocated to the intermembrane space of mitochondria, where it interacts with cytochrome c to produce ROS, such as hydrogen peroxide, and these p66shc-induced mitochondrial ROS stimulate oxidative stress-induced apoptosis [15,16]. On localization and activation in mitochondria, p66shc also participates in regulation of mitochondrial homeostasis, apoptosis, and cellular metabolic tone, as well as autophagosome formation [17,18]. In addition, in neuronal cells, knockdown of p66shc expression increased cell survival, whereas p66shc overexpression increased cell death involved in mitochondrial dysfunction, release of cytochrome c, and activation of the caspase cascade [19,20,21,22].

A recent study indicated that p66shc expression was significantly upregulated in the ipsilateral spinal cord in a chronic constriction injury model that produced neuropathic pain [23]. However, the roles of p66shc have not been fully elucidated. Therefore, the present study was performed to investigate the role of p66shc by introducing p66shc siRNA into a model of neuropathic pain induced by spinal nerve ligation (SNL) in rats. To deliver p66shc siRNA into spinal cord effectively and to reduce the number of treatment with p66shc siRNA, we performed the intrathecal administration with poly(d,l-lactic-*co*-glycolic acid) (PLGA) nanoparticles (NPs) containing p66shc siRNA (p66shc siRNA-PLGA NPs) because PLGA-based drug products are designed to reduce dosing frequency and potential drug toxicity, and siRNA-loaded NPs for gene therapy in various neurological diseases have attracted a great deal of attention [24,25,26]. In addition, to verify the role of p66shc, we also examined the effects of p66shc siRNA on H_2_O_2_-induced oxidative stress in HT22 neuronal cells.

## 2. Materials and Methods

### 2.1. Preparation, Characterization, Release Assessment, and Encapsulation Efficiency of p66shc siRNA-Encapsulated PLGA NPs

Fabrication of p66shc siRNA-PLGA NPs: We used PLGA nanoparticles (NPs) as the siRNA delivery system. p66shc siRNA-PLGA NPs were prepared from a PLGA nanoparticle synthesis service from the Nanoglia company (Daejeon, Korea). This method was previously reported by our team [27,28]. PLGA NPs with a 50:50 ratio of lactic and glycolic acid carrying p66shc siRNA (SHC1HSS, catalog no. RSS373234; Thermo Fisher, Waltham, MA, USA) or scrambled siRNA (product no. 12935400; Thermo Fisher) were prepared using an emulsification/solvent evaporation method. To produce p66shc siRNA-encapsulated PLGA NPs, 200 μL of 200 μM siRNA of a Tris-EDTA 8.0 buffer was added in drops to 800 μL of dichloromethane (DCM) (99.8%, D1641, Samchun Chem., Seoul, Korea) that contained 25 mg of PLGA (Purasorb, PDLG 5002A, Corbion, Amsterdam, The Netherlands) and then emulsified by sonication into a primary W1/O emulsion. Later, 2 mL of 1% polyvinyl alcohol (PVA) (98–99% hydrolyzed, Alfa Aeasar, Fisher Scientific) was directly added into the primary emulsion and further emulsified by sonication for 1 min to form a W1/O/W2 double emulsion. The resulting product was then diluted with 6 mL of 1% PVA and stirred magnetically for 3 h at room temperature to evaporate the DCM in a fume hood. Finally, the PLGA NPs were collected by centrifugation at 15,000× *g* for 15 min at 4 °C, washed twice with deionized water, and then divided into 10 vials and freeze-dried for 24 h. In 1 vial, the concentration of siRNA 20 μM and PLGA 2.5 mg is contained.

Characterization of p66shc siRNA-PLGA NPs: As described in our previous study [29], 20 mg of lyophilized particles were dispersed in 1 mL of deionized water to determine the size distribution and the zeta potential with Zetasizer Nano ZS (Malvern Instruments, Malvern, UK). The diameter and shape were also determined with scanning electron microscopy (SEM, SNE-4500 M, SEC Co., Suwon, Korea).

Release assessment of p66shc siRNA-PLGA NPs: We performed the release assessment of p66shc siRNA-PLGA NPs, according to the method of the previous studies [29,30]. Briefly, each 20 μM of siRNA-encapsulated PLGA NPs were incubated in 250 μL phosphate buffered saline (PBS, pH 7.4) at 37 °C, as a model of biological body fluids, for 48 h. At the designated time ((0, 3, 6, 9, 12, 24, 36 and 48 h after incubation), 200 μL of supernatant was collected, and the same amount of PBS was added again. The released p66shc siRNA was measured using NanoDrop (Thermo Fisher Scientific). 

Encapsulation efficiency of p66shc siRNA-PLGA NPs: The encapsulation efficiency of p66shc siRNA-PLGA NPs was evaluated according to the method of the previous studies [29,31]. The percentage of encapsulation efficiency was calculated as the ratio between the amount of siRNA released from the PLGA NPs and the amount of siRNA initially taken to prepare the PLGA NPs.

### 2.2. Experimental Animals

All animal-related procedures were performed in accordance with the guidelines of the Institutional Animal Care and Use Committee of Chungnam National University (CNUH-017-P0018 and CNUH-017-P0032) in compliance with National Institutes of Health regulations. Male Sprague–Dawley rats (180–200 g of weight) were purchased from Damul Science (Daejeon, Korea). Animals were housed individually in cages under a standard 12:12 h light:dark cycle at 25 ± 1 °C and water and food were available ad libitum. 

### 2.3. Neuropathic Pain Model, Intrathecal Injection of p66shc siRNA-Encapsulated PLGA NPs, and Behavioral Test

SNL-induced neuropathic pain model: According to the method of our previous studies [27,28,32], the induction of neuropathic pain was performed by SNL. In brief, the rats were anesthetized with 2% isoflurane (Hana Pharm Co., Seoul, Korea) in oxygen, and the L4 and L5 nerves were exposed after skin incision. Then, only the L5 nerve was separated from the exposed L4 and L5 nerves, and tightly ligated 3 times with the 3–0 silk thread for inducing neuropathic pain. The sham group did the same surgical procedure except for the L5 nerve ligation. After surgery, the surgical site was disinfected with povidone–iodine, and then the animals are kept in a home cage, confirming that it awakened from anesthesia. The process of observing whether the sensitive rats attacked each other due to surgery was performed for 3 days after surgery.

Intrathecal injection of p66shc siRNA-encapsulated PLGA NPs: At 3 days after SNL surgery, each 20 μM of siRNA-encapsulated PLGA NPs in 1 vial were prepared with 250 μL PBS, and the rats (*n* = 6 in each group) were anesthetized with 2% isoflurane in 1 L/min of oxygen, and then 20 μL of p66shc siRNA- or scrambled siRNA-encapsulated PLGA NPs in PBS (10 mg/mL, 1600 nM siRNA) was intrathecally administered between the L5–L6 intervertebral spaces using 25 μL syringes (26.5 G needle, Hamilton, Reno, NV, USA). 

Mechanical threshold test: SNL-induced pain behavior was measured by 0.16 to 15 g von Frey filaments (Stoelting Co., Wood Dale, IL, USA), according to the method of our previous studies [27,28,32]. Rats were allowed to acclimate 20 min on a 50 cm high mesh. Rats reacting at 10 g or more and 15 g or less were used in the experiment. Rat paws were stimulated 10 times at 5 min intervals with filaments, stimulation was given in 1-down force if the pain response was 5 or more, and stimulated in 1-up force if the pain response was less than 5 times. The rats, which responded with filaments ranging from less than 0.4 g on 3 days after surgery, were judged to have the SNL-induced neuropathic pain: the success rate of neuropathic pain modeling was about 80%. SNL-induced pain behavior was assessed by two investigators who did not perform the surgery as well as the intrathecal injection.

### 2.4. Western Blot Analysis

According to the method of our previous studies [28,32], Western blot analysis was performed. In brief, the ipsilateral dorsal horns of the L5 spinal cord region were collected and used for the experiment. After homogenizing the tissue in PRO-PREPTM (iNtRON Biotechnology, 17081, Kirkland, Washington, USA) and PNPP (Sigma, N9389, St. Louis, MO, USA), lysates were centrifuged. Protein was quantified by Bradford assay (BioRad, # 5000002, Hercules, CA, USA) using only supernatant and 20 μg of protein were used. Proteins were transferred using NC membrane and all blots were blocked with 5% skim milk (BD Biosciences, 90002-594, Franklin Lakes, NJ, USA). p-p66shc (Enzo Life Sciences, ALX-B04-358-C100, 1:1000, Lausen, Switzerland), total shc (BD Biosciences, #6108780, 1:1000), cleaved Caspase-3 (Cell signaling, 9661T, 1:1000), p62 (Sigma, P0067, 1:1000) and PINK1 (Novus Biologicals, NBP2-36488, 1:1000, Centennial, CO, USA) were used as primary antibodies. Visual confirmation of the band was developed using Clarity^TM^ Western ECL Substrate (BioRad, 170-5061).

### 2.5. Tissue Processing and Immunohistochemistry

According to the method of our previous studies [5,28], L5 spinal cord was obtained and immunohistochemical staining was performed. At 7 days after SNL, the rats (*n* = 6 in each group) were anesthetized by intraperitoneal administration with sodium pentobarbital (100 mg/kg) and perfused transcardially with 5% heparin diluted in 0.05 M PBS, followed by 4% paraformaldehyde (PFA, Millipore, 104005, Burlington, MA, USA) dissolved in 0.05 M PBS. Thereafter, the L5 spinal cord was obtained immediately, post-fixed in 4% PFA at 4 °C overnight, and cryoprotected by infiltration with 10% to 30% sucrose. The L5 spinal cord was serially sectioned to 30 μm thickness and immunostained with Iba-1 (Wako, 019-19741, 1:500, Osaka, Japan) as primary antibody. 

### 2.6. Cell Cultures 

The HT22 neuronal cell line was grown in DMEM/Dulbecco’s Modified Eagles Medium (High glucose) (HyClone, SH30243, Marlborough, MA, USA) and 10% Fetal Bovine Serum (HyClone, SH30084.03). After 6 h before transfection, the media was changed to Opti-MEM™ (Gibco™, 11058021, Waltham, MA, USA), and p66shc siRNA (SHC1HSS, ThermoFisher, Waltham, MA, USA) and scrambled siRNA (scRNA, ThermoFisher) were transfected with Lipofectamine™ 2000 Transfection Reagent (ThermoFisher, 11668027) and the concentration was treated with 100 pmol for 48 h. After removing the transfection medium, the cells were treated with DMEM media not containing serum diluted to a concentration of 100 μM of hydrogen peroxide solution (Sigma, H1009) for 3 h. The cells were then washed with PBS buffer and used for Western blot analysis and qPCR.

### 2.7. Quantitative Polymerase Chain Reaction (qPCR)

RNA extraction was performed using TRIzol™ Reagent (Life Technologies, 15596018), and cDNA was synthesized using the TOPscript™ RT DryMIX (Enzynomics, RT200) at a RNA concentration of 4 μg. qPCR was performed under the following conditions: 95 °C for 10 min, followed by 40 cycles of 95 °C for 15 s and 60 °C for 1 min using the AriaMx Real-time PCR System (Agilent Technologies, Santa Clara, CA, USA). Primer sequences (Cosmogenetech, Seoul, Korea), used in the qPCR, were as follows: mouse GAPDH, forward: 5′-ACC CAG AAG ACT GTG GAT GG-3′, reverse: 5′-CAC ATT GGG GGT AGG AAC AC-3′; mouse TNF-α (tumor necrosis factor-α), forward: 5′-AGC AAA CCA CCA AGT GGA GGA-3′, reverse: 5′-GCT GGC ACC ACT AGT TGG TTG T - 3′; mouse IL-1β (interleukin-1β), forward: 5′-TTG TGG CTG TGG AGA AGC TGT-3′, reverse: 5′-AAC GTC ACA CAC CAG CAG GTT-3′; mouse IL-6 (interleukin-6), forward: 5′-TCC ATC CAG TTG CCT TCT TGG-3′, reverse: 5′-CCA CGA TTT CCC AGA GAA CAT G-3′; mouse COX-2 (cyclooxygenase-2), forward: 5′-TGA GTA CCG CAA ACG CTT CT-3′, reverse: 5′-CTC CCC AAA GAT AGC ATC TGG-3′; mouse iNOS (inducible nitric oxide synthase), forward: 5′-GGC AAA CCC AAG GTC TAC GTT-3′, reverse: 5′-TCG CTC AAG TTC AGC TTG GT-3′; mouse p66shc, forward: 5′-GAC GAT AGT CCG ACT ACC CTG TGT-3′, reverse: 5′-CAC GAT AGT CCG ACT ACC CTG TGT-3′. The mRNA levels of each target gene were normalized to mRNA levels of GAPDH, and fold change was calculated using the 2^−ΔΔCt^ method.

### 2.8. Statistical Analysis

All statistical analyses were performed using GraphPad Prism 6.0 software (GraphPad Software Inc., La Jolla, CA, USA). The error values of the data are shown in the mean ± standard error of the mean (SEM). Differences among multiple groups were determined by one- or two-way analysis of variance (ANOVA) followed by an appropriate multiple comparison test. Between-group analyses were accomplished using unpaired Student’s *t*-tests. *P*-values 0.05 were considered statistically significant.

## 3. Results

### 3.1. Characterization of p66shc siRNA PLGA NPs

p66shc siRNA-PLGA NPs were prepared by sonication using the W/O/W double emulsion method to encapsulate hydrophilic siRNA. The average size and zeta potential were 183.7 ± 72.21 nm and 41.1 ± 4.81 mV, respectively (Figure 1A,B). Scanning electron microscopy showed that both p66shc siRNA-PLGA NPs and scrambled siRNA (scRNA)-PLGA NP were spheroids (Figure 1C). A siRNA release assay of NPs was conducted in PBS, and the p66shc siRNA-PLGA NPs showed relatively a stable release for 48 h: the cumulative siRNA release was 96.4% within 48 h (Figure 1D). In addition, we found that the encapsulation efficiency of p66shc siRNA-PLGA NPs was 32.3%.

### 3.2. Effect of p66shc siRNA-PLGA NPs on the SNL-Induced Mechanical Hypersensitivity.

To evaluate the analgesic effects of p66shc siRNA-PLGA NPs on the mechanical hypersensitivity following SNL, we carried out intrathecal administration of p66shc siRNA-PLGA NPs or scRNA-PLGA NPs three days after SNL (Figure 2A), and measured the paw withdrawal threshold at 3, 5, 7, 9, and 14 days after SNL (Figure 2B). No significant changes in mechanical threshold allodynia were observed in the sham group. However, the mechanical threshold in the scrambled siRNA-PLGA NP treatment group after SNL (SNL group) was decreased markedly from 3 to 14 days following SNL, compared with that in the sham group. On the other hand, the mechanical threshold in the p66shc siRNA-PLGA NP treatment group after SNL (p66shc siRNA-SNL group) was much higher than that in the SNL group, especially at 7 and 9 days after SNL (Figure 2B).

Based on these results, we performed further experiments to elucidate the mechanisms underlying the analgesic effects of p66shc siRNA-PLGA NPs at seven days after SNL using the L5 spinal cord.

### 3.3. Changes in p66shc and Markers for Apoptosis, Autophagy, and Mitophagy Following SNL

Next, we investigated whether the SNL-induced mechanical allodynia was attributed to the phosphorylation of p66shc, which has been shown to be associated with mitochondrial dysfunction [22]. In the ipsilateral dorsal horn of the L5 spinal cord at seven days after SNL, the levels of phosphorylated (p)-p66shc and total shc were markedly increased in the SNL group compared with the sham group. However, the p-p66shc, but not total shc, level was significantly lower in the p66shc siRNA-SNL group than the SNL group (Figure 3A,B).

We also evaluated the changes in levels of cleaved caspase-3 (marker of apoptosis), p62 (marker of autophagy), and PINK1 (marker of mitophagy) in the L5 spinal cord after SNL. We found the SNL-induced significant increases of cleaved caspase-3, p62, and PINK1 levels in the SNL group. However, in the p66shc siRNA-SNL group, the cleaved caspase-3 level was decreased by 19.1% ± 0.24% compared with that in the SNL group. In addition, p62 and PINK1 levels in the p66shc siRNA-SNL group were also decreased by 38.3% ± 0.03% and 74.2% ± 0.09%, respectively, compared with those in the SNL group (Figure 3C,D).

### 3.4. Activation of Microglia in the Ipsilateral Spinal Cord Following SNL

It has been well known that microglia were proliferated and activated in the spinal cord, and that activated microglia could mediate the neuroinflammatory process in the spinal cord, which contribute to the development and maintenance of neuropathic pain [27,28,32,33]. Therefore, in this study, we examined the activation of microglia in the ipsilateral dorsal horn of the L5 spinal cord following SNL. In the SNL group, the number of Iba-1-immunoreactive microglia was markedly increased in the ipsilateral dorsal horn of the L5 spinal cord at seven days after SNL. On the other hand, the number of Iba-1-immunoreactive microglia was significantly decreased in the p66shc siRNA-SNL group compared with the SNL group (Figure 4).

### 3.5. Reduced Protein Levels of p-p66shc and Markers of Apoptosis, Autophagy and Mitophagy by p66shc siRNA in H_2_O_2_-Stimulated HT22 Cells

Next, to verify the role of p66shc, we examined whether p66shc siRNA attenuates the expression of markers of apoptosis, autophagy, and mitophagy in H_2_O_2_-stimulated HT22 cells. First, we confirmed that the marked increase in the p-p66shc level in the scRNA+H_2_O_2_ group significantly decreased by p66shc siRNA treatment: p-p66shc level of p66shc siRNA-treated group was similar to that of the scRNA-treated group (as control group) (Figure 5). In addition, we found that H_2_O_2_-mediated increases in cleaved caspase-3, p62, and PINK1 protein levels were significantly inhibited by p66shc siRNA treatment (Figure 5).

### 3.6. Suppression of Transcription of Proinflammatory Mediators by p66shc siRNA in H_2_O_2_-Stimulated HT22 Cells

We investigated whether p66shc siRNA inhibits the mRNA expression of proinflammatory mediators in H_2_O_2_-stimulated HT22 cells. The mRNA levels of proinflammatory mediators, including TNF-α, IL-1β, IL-6, COX2, and iNOS, were increased markedly in the scRNA+H_2_O_2_ group but were significantly decreased in the p66shc siRNA group by 38.5% ± 0.05%, 61.3% ± 0.05%, 44.0% ± 0.07%, 30.9% ± 0.18%, and 28.5% ± 0.15%, respectively, compared with the scRNA+H_2_O_2_ group (Figure 6).

## 4. Discussion

PLGA and NPs have been widely used as an established vehicle for siRNA delivery [24,26,27,28,29]. In addition, our previous studies showed that PLGA NPs were effectively absorbed in the spinal cord of rats with SNL-induced neuropathic pain: we found that the intrathecally administrated PLGA NPs were observed in neurons as well as in astrocytes and microglia in the L4–L6 dorsal horn of the spinal cord of rats [27,28]. Thus, in this study, we used PLGA NPs to deliver p66shc siRNA to elucidate the role of p66shc in SNL-induced neuropathic pain.

A recent study reported that p66shc in the ipsilateral spinal cord plays important roles in the development of neuropathic pain, and inhibition of p66shc by intrathecal administration of carnosic acid is closely involved in the anti-nociceptive effect [23]. In the present study, we found that the SNL-induced reduction in the mechanical threshold was partially recovered by intrathecal administration of p66shc siRNA-PLGA NPs at 7 and 9 days after SNL. This result suggested that downregulation of p66shc blocked the SNL-induced mechanical hypersensitivity on the ipsilateral hind paw and produced an analgesic effect. In addition, our observations were consistent with the results of a previous study indicating that no age-dependent increase in pain sensitivity was observed in p66shc-knockout mice compared with wild-type controls [34].

It has been reported that p66shc^−/−^ cells, as well as various tissues in p66shc-knockout mice, are resistant to apoptosis induced by various different signals and pathological states, and that neuronal apoptotic cascades, including the release of cytochrome C and activation of caspase-3, occur in SNL-induced neuropathic pain [19,35,36,37]. It was also reported that autophagy is closely associated with neuropathic pain following SNL, and that SNL-induced upregulation of autophagy markers, involved in abnormal accumulation of dysfunctional autophagosomes, occurred as a result of dysfunctional autophagic flux in SNL-induced neuropathic pain [10,12,13,38]. Berliocchi et al. [10] suggested that neuropathic pain after SNL is a consequence of impaired basal autophagy by blockade of autophagosome turnover. They showed that the p62 level was increased in the mouse spinal cord at 7 days after SNL and suggested that accumulation of p62 represented disruption of the autophagic process and autophagic flux [10]. In addition, our previous study showed that PINK1 expression was increased in the spinal dorsal horn neurons in SNL-induced neuropathic pain, and that PINK1-knockout mice exhibited reduced neuropathic hypersensitivity [32]. We suggest that aberrant mitophagic flux occurs in the SNL-induced neuropathic pain model, and that regulation of mitophagy may represent a therapeutic target in the management of neuropathic pain [32]. In the present study, we observed that the protein levels of cleaved caspase-3, p62, and PINK1 were significantly decreased in the p66shc siRNA-SNL group compared with the SNL group. In addition, we also found significant reductions in the levels of cleaved caspase-3, p62, and PINK1 by p66shc siRNA in H_2_O_2_-stimulated HT22 cells. Therefore, based on these results, it is likely that the partial analgesic effect of p66shc siRNA-PLGA NPs is due to reduced SNL-induced apoptosis as well as SNL-induced aberrant autophagic and mitophagic processes.

Neuropathic pain is closely associated with neuroinflammatory processes, including upregulated proinflammatory mediators, which may participate in the pain process [39,40]. It has also been reported that autophagy is intimately associated with modulation of neuroinflammatory processes, and inhibition of autophagy can reduce the miR-195-induced increases in neuroinflammation and neuropathic pain following SNL [41,42]. In addition, p66shc has been recognized as a proinflammatory molecule, and altered expression of p66shc in the spinal cord during neuropathic pain may participate in the modulation of proinflammatory cytokines [23,43]. In the present study, intrathecal administration of p66shc siRNA-PLGA NPs decreased microglial activation in the ipsilateral spinal cord following SNL. Our and others’ previous studies showed that microglia are activated distinctly, and that activated microglia lead to excessive neuroinflammation following peripheral nerve injury, which contributes to neuropathic pain and tissue damage [5,27,28,32,33,44]. In addition, in the present study, we showed that p66shc siRNA treatment decreased H_2_O_2_-induced elevation of the mRNA levels of proinflammatory mediators, including TNF-α, IL-1β, IL-6, COX2, and iNOS, in cultured HT22 cells. Therefore, decreased microglial activation as well as decreased production of proinflammatory mediators by p66shc downregulation may also be associated with anti-nociceptive effects in SNL-induced neuropathic pain.

In conclusion, the results of the present study suggest that intrathecal administration of p66shc siRNA-PLGA NPs, which led to the inhibition of p66shc expression in the spinal cord, could partially attenuate the development of SNL-induced neuropathic pain and exert an anti-nociceptive effect. These effects of p66shc downregulation by p66shc siRNA-PLGA NPs may be closely associated with the attenuations of the SNL-induced aberrant autophagic, mitophagic, and neuroinflammatory processes in rats.

## Figures and Tables

**Figure 1 polymers-12-01014-f001:**
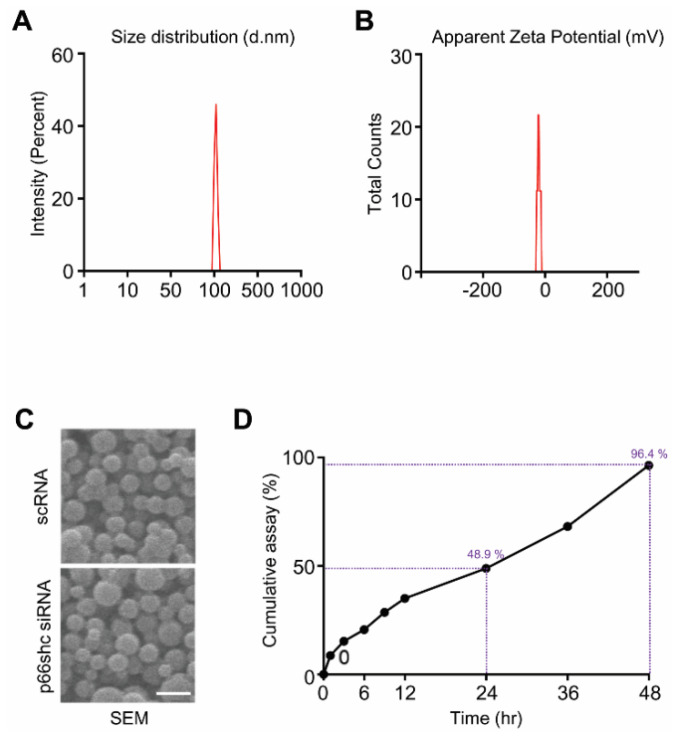
Characterization of p66shc siRNA-PLGA NPs. p66shc siRNA-PLGA NPs were prepared by briefly sonicating the PLGA mixture and then examined for (**A**) size (d.nm: diameter in nm); (**B**) zeta potential using the Zetasizer Nano ZS90; (**C**) by scanning electron microscopy; scale bar = 200 nm, and (**D**) the cumulative siRNA release from PLGA NPs.

**Figure 2 polymers-12-01014-f002:**
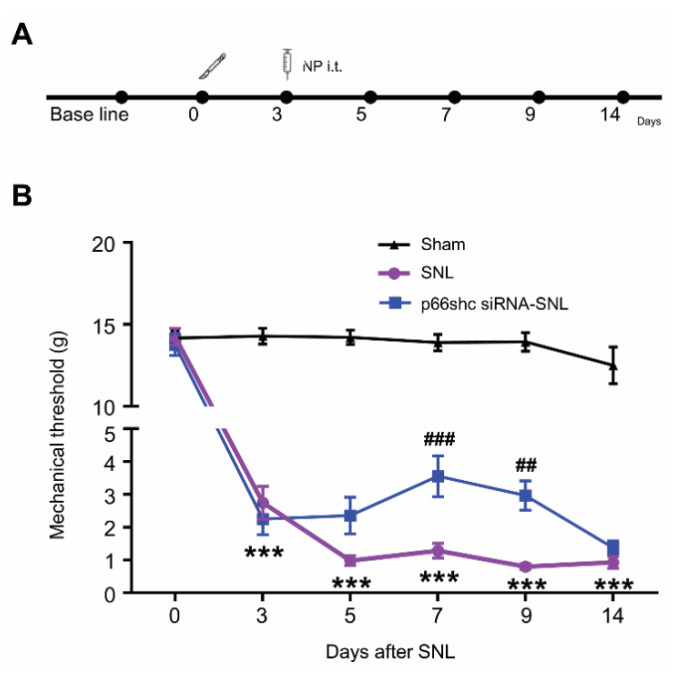
The degree of pain confirmed by the mechanical threshold after spinal nerve ligation (SNL). (**A**) experimental timeline of this study; (**B**) After three days of SNL-induced neuropathic pain, the mechanical threshold was identified using the von Frey filament test to confirm the analgesic effect of p66shc siRNA-PLGA NPs (*n* = 6 in each group). Data are expressed as means ± SEM (two-way ANOVA with Dunnett’s post hoc test, *** *p* < 0.001 vs. sham group, ^###^
*p* < 0.001, ^##^
*p* < 0.01 vs. SNL group.

**Figure 3 polymers-12-01014-f003:**
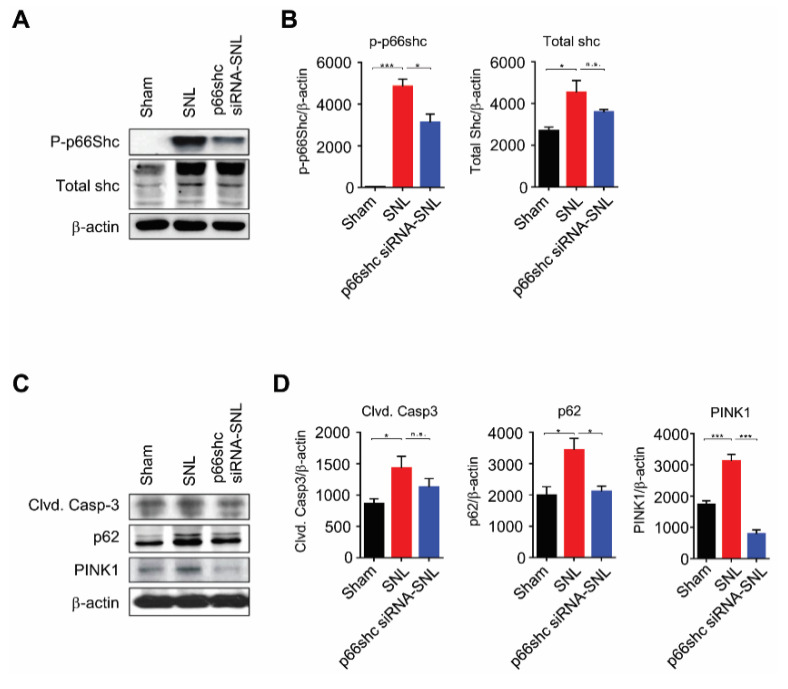
p66shc siRNA-PLGA NPs reduced the protein levels of cleaved caspase-3, p62, and PINK1 in the ipsilateral spinal cord at 7 days after SNL. (**A**,**B**) phosphorylation of p66shc was decreased after treatment with p66shc siRNA-PLGA NPs; (**C**,**D**) SNL-induced increases in cleaved caspase-3, p62, and PINK1 protein levels were significantly attenuated in the p66shc siRNA-SNL group. β-actin was used as a loading control. Data are expressed as means ± SEM (unpaired Student’s *t*-test, *** *p* < 0.001, ** *p* < 0.01, * *p* < 0.05 vs. sham or SNL group).

**Figure 4 polymers-12-01014-f004:**
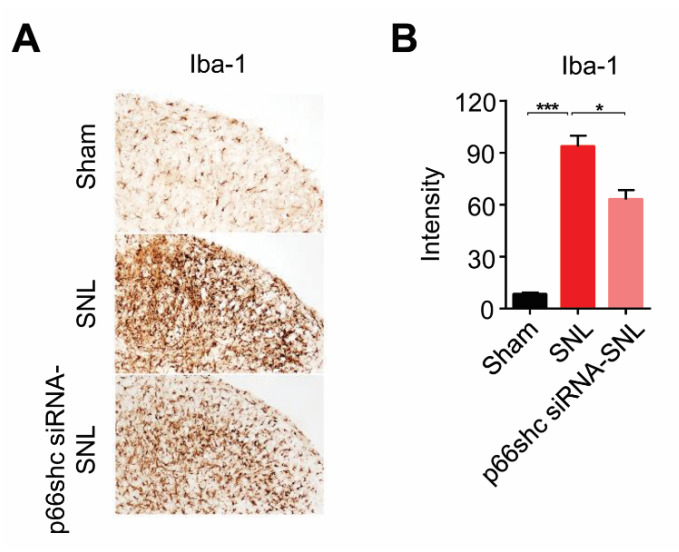
Iba-1 immunoreactivity in the ipsilateral spinal cord at seven days after SNL. (**A**) microglial activation was reduced in the p66shc siRNA-PLGA NPs group compared with the SNL group. Scale bar = 100 μm; (**B**) intensity of Iba-1 immunoreactivity in the ipsilateral spinal cord after SNL. Data are expressed as means ± SEM (unpaired Student’s *t*-test, *** *p* < 0.001 vs. sham, * *p* < 0.05 vs. SNL).

**Figure 5 polymers-12-01014-f005:**
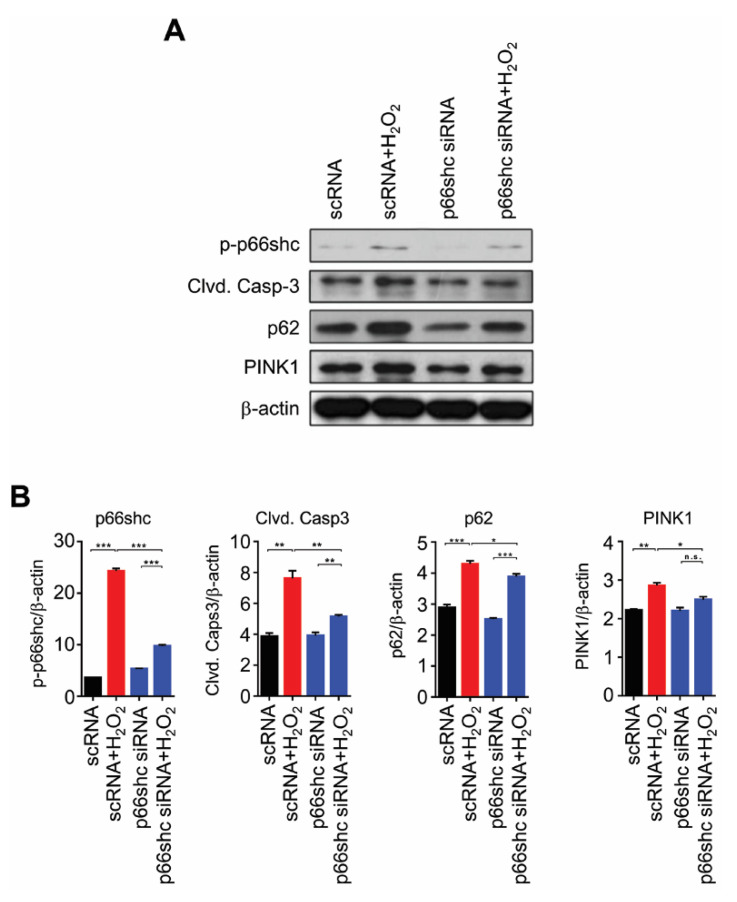
Reduced expression of markers of apoptosis, autophagy, and mitophagy after p66shc knockdown in H_2_O_2_-induced HT22 cells. (**A**) induction of oxidative stress by H_2_O_2_ after p66shc knockdown in the HT22 neuronal cell line reduced the protein levels of phosphorylated p66shc, cleaved caspase-3, p62, and PINK1; (**B**) β-actin was used as a loading control. Western blotting showed that the levels of p66shc, cleaved caspase-3, p62, and PINK1 were significantly altered in HT22 cells under oxidative stress induced by H_2_O_2_. Data are expressed as means ± SEM (unpaired Student’s *t*-test, *** *p* < 0.001, ** *p* < 0.01).

**Figure 6 polymers-12-01014-f006:**
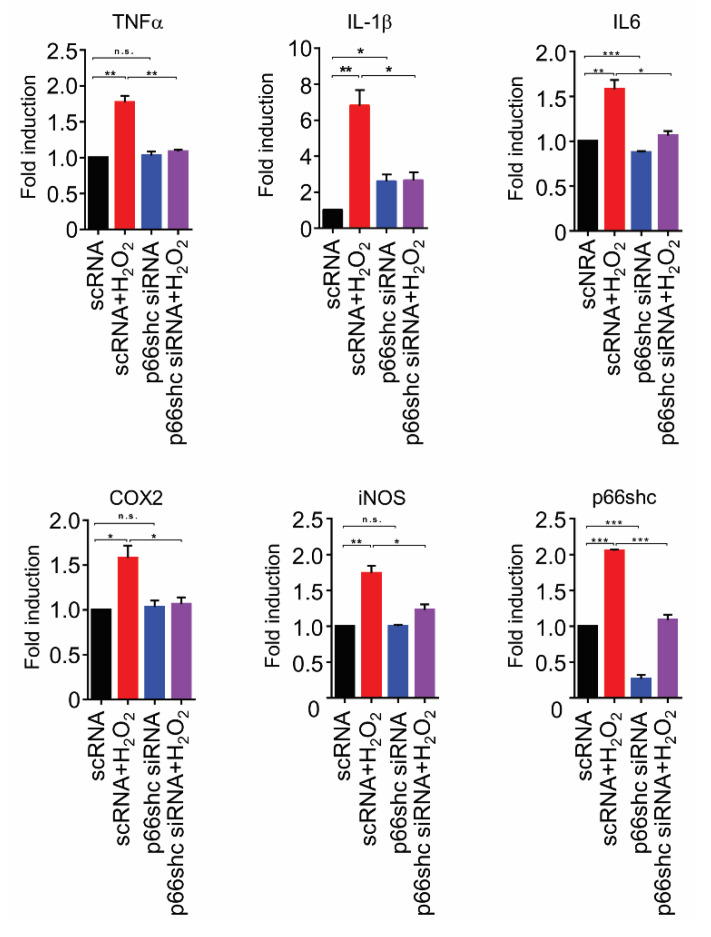
p66shc knockdown resulted in decreased mRNA levels of proinflammatory mediators in H_2_O_2_-induced HT22 cells. TNF-α: tumor necrosis factor-α, IL-1β: interleukin-1β, IL-6: interleukin-6, COX-2: cyclooxygenase-2, iNOS: inducible nitric oxide synthase. Data are expressed as means ± SEM (unpaired Student’s *t*-test, *** *p* < 0.001, ** *p* < 0.01, * *p* < 0.05).

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
