# Peer review of "p66shc siRNA-Encapsulated PLGA Nanoparticles Ameliorate Neuropathic Pain Following Spinal Nerve Ligation"

_polymers, 2020, doi:10.3390/polym12051014_

Round 1
Reviewer 1 Report
The manuscript titled “p66shc siRNA-encapsulated PLGA nanoparticles ameliorate neuropathic pain following spinal nerve ligation”, which performed to investigate whether p66shc siRNA-encapsulated poly(D,L-lactic-co-glycolic acid) nanoparticles (p66shc siRNA-PLGA NPs) can attenuate spinal nerve ligation (SNL)-induced neuropathic pain. Moreover, the effects of p66shc siRNA on H2O2-induced oxidative stress in HT22 neuronal cells were examined to elucidate the role of p66shc. In my opinion, it is not well written, and no uses standard methods. The experimental data were not sufficient. Several deficits should be noted:
- The manuscript mentions “The cumulative siRNA release was 96.4% within 48 hours”, so why is the data not shown in this manuscript? I think it necessary to show this.
- Why not introduce the source of PLGA and other chemical reagents?
- The manuscript mentions “In addition, our previous study showed that PLGA NPs were effectively absorbed in the spinal cord of rats with SNL-induced neuropathic pain”, so how did that PLGA NPs effectively absorbed in the spinal cord of rats with SNL-induced neuropathic pain? Why is there no explanation here? I think it necessary to explain this in this manuscript.
- Why is there no detailed explanation of how p66shc siRNA was released from the NPs in the manuscript? There should be a supplementary explanation in the manuscript about this.
- The manuscript mentions “To deliver p66shc siRNA, we used poly(D,L-lactic-co-glycolic acid) (PLGA) nanoparticles (NPs) containing p66shc siRNA (p66shc siRNA-PLGA NPs), because PLGA-based drug products are designed to reduce dosing frequency and potential drug toxicity”, so why PLGA nanoparticles (NPs) containing p66shc siRNA (p66shc siRNA-PLGA NPs) work better? Additional experiments should be performed to text drug toxicity about p66shc siRNA and p66shc siRNA-PLGA NPs.
Author Response
Manuscript ID: Polymers-762381
Title: p66shc siRNA-encapsulated PLGA nanoparticles ameliorate neuropathic pain following spinal nerve ligation
Dear Reviewer
I appreciate the reviewer’s prudent comments on our manuscript. The responses to the reviewer’ comments are summarized below:
Response to Reviewer’s Comment:
Comment 1: The manuscript mentions “The cumulative siRNA release was 96.4% within 48 hours”, so why is the data not shown in this manuscript? I think it necessary to show this.
Response: According to the reviewer’s comment, we represented the graph for cumulative siRNA release from PLGA NPs in the new Figure 1D. Please refer to the new Figure 1.
Comment 2: Why not introduce the source of PLGA and other chemical reagents?
Response: According to the reviewer’s comment, we described the source of PLGA and other chemical reagents. Please refer to the text.
Comment 3: The manuscript mentions “In addition, our previous study showed that PLGA NPs were effectively absorbed in the spinal cord of rats with SNL-induced neuropathic pain”, so how did that PLGA NPs effectively absorbed in the spinal cord of rats with SNL-induced neuropathic pain? Why is there no explanation here? I think it necessary to explain this in this manuscript.
Response: In our previous studies, we found that the intrathecally administrated PLGA NPs were observed in neurons as well as in astrocytes and microglia in the L5 dorsal horn of the spinal cord of rats (Shin et al. Nanomedicine(Lond) 2018, 13:1607-1621; Shin et al. Nanomedicine 2019, 18:90-100). According to the reviewer’s comment, we described this. Please refer to the text.
Comment 4: Why is there no detailed explanation of how p66shc siRNA was released from the NPs in the manuscript? There should be a supplementary explanation in the manuscript about this.
Response: As the reviewer knows, the degradation mechanism of PLGA is a complex matter in and of itself. Even if it has been studied since the 1960s and 1970s and huge advances have been made in the understanding of in vitro degradation mechanisms a complete understanding has yet to be achieved, since many factors affect polymer degradation and the consequent active pharmaceutical ingredient (API) release (Paolo Blasi, J Pharm Investig 2019, 49: 337-346). PLGA co-polymers undergo degradation in the presence of water, both in vitro and in vivo. The ester bonds are cleaved by hydrolytic degradation occurring throughout the whole PLGA microparticle matrix. PLGA degradation can be divided into three phases. In the first phase of random chain scission, the molecular weight of the polymer decreases significantly while the weight loss and soluble monomer formed are not appreciable. During the middle phase there is further decrease of the molecular weight and rapid loss of mass, with the production of soluble oligomers and monomers. In the last step the polymer disappears completely, because of the formation of soluble monomers. Once the monomers are formed, they are eliminated by physiological pathways. Lactic acid enters the tricarboxylic acid cycle and is metabolized and eliminated in carbon dioxide and water, while glycolic acid is excreted unchanged by the kidneys or metabolized by the tricarboxylic acid cycle (Anderson and Shive, Adv Drug Delivery Rev 2012, 64: 72-82; Elmowafy et al., J Pharm Investig 2019, 49: 347-380; Paolo Blasi, J Pharm Investig 2019, 49: 337-346).
However, as the reviewer knows, it has been generally and widely accepted that the drug loading and release rates from PLGA particles do not necessarily conform to predicted behavior as the effect of microparticle size on drug release kinetics quantitatively can only be predicted for certain well-defined formulations. (Han et al., Front Pharmacol 2016, 7: 185). Therefore, based on this previous study, it is hard to describe a detail explanation of how p66shc siRNA was released from the NPs in the manuscript, although the reviewer pointed out. However, as responded in the comment 1, we found that the p66shc siRNA-PLGA NPs showed relatively a stable release for 48 hours based on the result of a siRNA release assay. Please comprehend this and refer to the text.
Comment 5: The manuscript mentions “To deliver p66shc siRNA, we used poly(D,L-lactic-co-glycolic acid) (PLGA) nanoparticles (NPs) containing p66shc siRNA (p66shc siRNA-PLGA NPs), because PLGA-based drug products are designed to reduce dosing frequency and potential drug toxicity”, so why PLGA nanoparticles (NPs) containing p66shc siRNA (p66shc siRNA-PLGA NPs) work better? Additional experiments should be performed to text drug toxicity about p66shc siRNA and p66shc siRNA-PLGA NPs.
Response: As the reviewer knows, 15 FDA-approved PLA/PLGA-based drug products are available on the US market, and PLA/PLGA-based drug products are designed to reduce dosing frequency and potential drug toxicity (Jain et al. Adv Drug Deliv Rev 2016, 107:213-227). In addition, it was reported that the flavopiridol-encapsulated PLGA NPs, which delivered topically in the spinal cord, showed relatively stable release for 3 days, and blank PLGA NPs did not show any cytotoxicity (Ren et al. Biomaterials 2014, 35:6585-6594). They suggested that PLGA NPs have considerable safety profiles and that PLGA NPs can be an appropriate carrier for therapeutic molecules (Ren et al. Biomaterials 2014, 35:6585-6594). In our previous studies, we also performed MTT assay, and we reported that there was no any cytotoxicity of p38 siRNA-encapsulated PLGA NPs or plasmids-encapsulated PLGA NPs (Shin et al. Nanomedicine(Lond) 2018, 13:1607-1621; Shin et al. Nanomedicine 2019, 18:90-100). Therefore, based on the results of above our and other previous studies, we thought that p66shc siRNA-PLGA NPs have little toxicity, although we did not perform a toxicity test in this study. Please comprehend this and our opinion.
We deeply thank again for the prudent comments on our manuscript.
Very sincerely yours,
Corresponding Author:
Choong-Hyun Lee, DVM, PhD.
Department of Pharmacy, College of Pharmacy, Dankook University, Cheonan, 31116, Republic of Korea.
TEL: +82-41-550-1441. FAX: +82-41-559-7899.
E-mail: [email protected]

Reviewer 2 Report
Authors describe “p66shc siRNA-encapsulated PLGA nanoparticles ameliorate neuropathic pain following spinal nerve ligation” Some closely related work is also reported, authors have to provide justification and importance of this work. It needs some modifications as well as suggested below:
Comments
1 Author work is closely related with the work reported by Shin, J. et al. p38 siRNA-encapsulated PLGA nanoparticles alleviate neuropathic pain behavior in rats by inhibiting microglia activation. Nanomedicine (Lond) 2018, 13, 1607-1621. The delivery material is also same. Why author think that this work should be published and what is research gap between this work and previous work?
2 Experimental section should be rearranged, first should be preparation of PLGA, encapsulation then animal studies
3 Since the authors are doing in vivo studies, What about the cytotoxicity of PLGA? 4 What about stability of PLGA nanoparticles?
5 What are the selection criteria of PLGA in this study as delivery material?
6 What is the role of PVA in nanoparticles preparation? How author think that 1 % PVA is enough amount?
7 What is the % encapsulation of siRNA in PLGA nanoparticles and how it was calculated? Authors didn’t provide data about % encapsulation; it should be explained and discussed.
8 A graph for siRNA release from PLGA should be provided in the supplementary material, also how it was calculated.
9 Figure 1, a,b, c are not clear, please provide clear images with good resolution. The size from SEM is also not clear.
10 Also improve resolution of all figures.
11 Results need to elaborate further.
12 Aim of the study should be clearly elucidated in the introduction.
13 Except few, most of the references are old. Please cite current references.
14 Level of English should be improved further.
15 Abstract and conclusion should be improved further.
16 Plagiarism of manuscript is high, please revise it carefully especially experimental part.

Author Response
Manuscript ID: Polymers-762381
Title: p66shc siRNA-encapsulated PLGA nanoparticles ameliorate neuropathic pain following spinal nerve ligation
Dear Reviewer
I appreciate the reviewer’s prudent comments on our manuscript. The responses to the reviewer’ comments are summarized below:
Response to Reviewer’s Comment:
Comment 1: Author work is closely related with the work reported by Shin, J. et al. p38 siRNA-encapsulated PLGA nanoparticles alleviate neuropathic pain behavior in rats by inhibiting microglia activation. Nanomedicine (Lond) 2018, 13, 1607-1621. The delivery material is also same. Why author think that this work should be published and what is research gap between this work and previous work?
Response: As the reviewer mentioned, our present study is closely related with the previous study using same delivery material (PLGA nanoparticles) (Shin et al. Nanomedicine(Lond) 2018, 13:1607-1621). As the reviewer knows, PLA/PLGA-based drug products are designed to reduce dosing frequency and potential drug toxicity, and it has been widely accepted that PLGA nanoparticles have considerable safety profiles and that PLGA NPs can be an appropriate carrier for therapeutic molecules (Jain et al. Adv Drug Deliv Rev 2016, 107:213-227; Ren et al. Biomaterials 2014, 35:6585-6594). Therefore, we used PLGA nanoparticles as a delivery material in both present and previous studies.
As the reviewer knows, many underlying mechanisms, related to the onset and development of neuropathic pain, has been suggested such as alterations in neuronal activities, immune responses, neuroinflammatory processes, oxidative stress, as well as mitochondrial dysfunction. However, the effectiveness of pharmacological treatment for neuropathic pain remains limited due to lots of etiologies. Thus, many other studies have shown the changes in a large number of molecules in the neuropathic pain, and suggested the numerous therapeutic targets for neuropathic pain.
It has been well known that the phosphorylation of p38 MAPK results in the production of proinflammatory mediators, resulting in increased mechanical hypersensitivity and aggravation of pain symptoms, and that the inhibition of the p38 in the spinal microglia could be capable of reducing neurologic pain. Therefore, in our previous study, we focused and examined the relationships between p38 and neuroinflammatory processes in the neuropathic pain, and we showed that p38 siRNA NPs significantly reduced mechanical allodynia as well as microgliosis in the spinal dorsal horns of SNL rats, consistent with a downregulation of proinflammatory mediators (Shin et al. Nanomedicine(Lond) 2018, 13:1607-1621). On the other hand, in the present study, we focused the role of p66shc in the neurologic pain, because p66shc has diverse roles in regulation of mitochondrial homeostasis, apoptosis, as well as autophagosome formation, and we represented that p66shc was associated with the development of SNL-induced neuropathic pain as well as the antinociceptive effect, and that effects of p66shc downregulation may be related with attenuation of aberrant autophagic and mitophagic as well as neuroinflammatory processes following SNL. Therefore, based on the results of our previous and present studies, we suggest that the regulations of p38 and/or p66shc could be one of new therapeutic targets for the treatment of neuropathic pain. In addition, we think that it is so meaningful to investigate the role of p66shc by introducing PLGA nanoparticles containing p66shc siRNA into a model of neuropathic pain induced by spinal nerve ligation (SNL) in rats. Please comprehend this and our opinion.
Comment 2: Experimental section should be rearranged, first should be preparation of PLGA, encapsulation then animal studies
Response: According to the reviewer’s comment, we rearranged it. Please refer to the text.
Comment 3: Since the authors are doing in vivo studies, What about the cytotoxicity of PLGA?
Response: As responded in the Comment 1, PLA/PLGA-based drug products are designed to reduce dosing frequency and potential drug toxicity, and it has been widely accepted that PLGA nanoparticles have considerable safety profiles and that PLGA NPs can be an appropriate carrier for therapeutic molecules (Jain et al. Adv Drug Deliv Rev 2016, 107:213-227; Ren et al. Biomaterials 2014, 35:6585-6594). Ren et al. (2014) showed that blank PLGA NPs did not show any cytotoxicity. In addition, in our previous studies, we also performed MTT assay, and we reported that there was no any cytotoxicity of p38 siRNA-encapsulated PLGA NPs or plasmids-encapsulated PLGA NPs (Shin et al. Nanomedicine(Lond) 2018, 13:1607-1621; Shin et al. Nanomedicine 2019, 18:90-100). Therefore, based on the results of above our and other previous studies, we thought that p66shc siRNA-PLGA NPs have little toxicity, although we did not perform a toxicity test in this study. Please comprehend this and our opinion.
Comment 4: What about stability of PLGA nanoparticles?
Response: As the reviewer knows, physical properties of PLGA have been shown to depend on different factors, including the initial molecular weight of the monomers, the LA:GA ratio, the exposure time to water and the storage temperature. The table of paper show the physico-chemical properties and the field of application of different PLGA materials characterized by different LA:GA ratio (Gentile et al., 2014, Int J Mol Sci 15, 3640-3659). They showed that PLGA nanoparticle, which our lab used, was 2.0 Modulus and 3-10% elongation and its degradation time is 1-2 weeks. In addition, FDA has approved use of PLGA in many human anticancer and anti-viral therapeutics (Kumari et al. 2010, Colloids Surf B Biointerfaces, 75(1):1-18). 9-Nitrocamptothecin (9-NC) is an antitumor therapeutic that can be encapsulated by PLGA via nanoprecipitation to prevent pH dependent instability. Encapsulated 9-NC has been shown to work well in vitro and has demonstrated a sustained release up to 160 hours. Paclitaxel, a drug which causes cell death by polymerizing tubulin, is used to treat ovarian and colon cancers, and can be encapsulated by PLGA via interfacial deposition method to increase. One group found that encapsulating Paclitaxel increased efficiency of the drug in vitro via enhancing the drug’s cytotoxic effect. Furthermore they found that these PLGA nanoparticles release Paclitaxel rapidly in the first 24 hours, then more gradually over the next few days (Fonseca et al. 2002, J Control Release 2002, 83(2):273-286). In addition, it was reported that the flavopiridol-encapsulated PLGA NPs, which delivered topically in the spinal cord, showed relatively stable release for 3 days, and blank PLGA NPs did not show any cytotoxicity (Ren et al. Biomaterials 2014, 35:6585-6594). They suggested that PLGA NPs have considerable safety profiles and that PLGA NPs can be an appropriate carrier for therapeutic molecules (Ren et al. Biomaterials 2014, 35:6585-6594). But, in this study, we did not evaluate the stability of PLGA nanoparticles. Although we could not examine the stability of PLGA nanoparticles with our present capacity, we observed that p66shc siRNA-PLGA nanoparticles showed relatively stable release for 48 hr, as responded in the comment 8 and represented in the new Figure 1. Please comprehend this.
Comment 5: What are the selection criteria of PLGA in this study as delivery material?
Response: As the reviewer knows, 15 FDA-approved PLA/PLGA-based drug products are available on the US market, and PLA/PLGA-based drug products are designed to reduce dosing frequency and potential drug toxicity, and it has been widely accepted that PLGA nanoparticles have considerable safety profiles and that PLGA NPs can be an appropriate carrier for therapeutic molecules (Jain et al. Adv Drug Deliv Rev 2016, 107:213-227; Ren et al. Biomaterials 2014, 35:6585-6594), as responded above. Therefore, on the basis of merit of PLGA, we have used PLGA as a delivery material in the present and previous studies. Please comprehend this.
Comment 6: What is the role of PVA in nanoparticles preparation? How author think that 1 % PVA is enough amount?
Response: As the reviewer knows, poly(vinyl alcohol) (PVA) has been widely and commonly used as an emulsifying agent in the preparation of biodegradable PLGA polymeric nanoparticles. In addition, many previous studies used the various concentrations (0.1 to 1 %) PVA for the preparation of PLGA nanoparticles (Shameem et al. AAPS PharmSci 1999, 1:E7; Wohlfart et al. PLoS One 2011, 6:e19121). In our previous study, we used 1 % PVA for the preparation of Foxp3-containing PLGA nanoparticles, and we found that the intrathecally injected Foxp3-containing PLGA nanoparticles were preferentially localized in the spinal cord (Shin et al. Nanomedicine 2019, 18:90-100). Therefore, we used 1 % PVA in the present study, and we think that 1 % PVA is enough amount. Please comprehend this and our opinion.
Comment 7: What is the % encapsulation of siRNA in PLGA nanoparticles and how it was calculated? Authors didn’t provide data about % encapsulation; it should be explained and discussed.
Response: In the present study, we measured the encapsulation efficiency of p66shc siRNA-PLGA NPs, according to the method of the previous studies (Shin et al. Polymers (Basel) 2020, 12(2) pii: E443, doi: 10.3390/polym12020443; Peltonen et al. AAPS PharmSciTech 2004, 5: E16). The percentage of encapsulation efficiency was calculated as the ratio between the amount of siRNA released from the PLGA NPs and the mount of siRNA initially taken to prepare the PLGA NPs, and we found that encapsulation efficiency was 32.3 %. Please refer to the text.
Comment 8: A graph for siRNA release from PLGA should be provided in the supplementary material, also how it was calculated.
Response: According to the reviewer’s comment, we represented the graph for cumulative siRNA release from PLGA NPs in the new Figure 1D, and described the method in the Materials and Methods section. Please refer to the text.
Comment 9: Figure 1, a,b, c are not clear, please provide clear images with good resolution. The size from SEM is also not clear.
Response: According to the reviewer’s comment, we improved resolution of Figure 1 and showed scale bar. Please refer to the new figure 1.
Comment 10: Also improve resolution of all figures.
Response: As responded in the comment 9, we improved resolution of all figures. Please refer to the new figures.
Comment 11: Results need to elaborate further.
Response: According to the reviewer’s comment, we checked the Result section again and rewrote some parts. Please refer to the text.
Comment 12: Aim of the study should be clearly elucidated in the introduction.
Response: The aim of the present study was to investigate the role of p66shc by introducing p66shc siRNA in the SNL-induced neuropathic pain. However, as the reviewer knows, a delivery of siRNA to the target remains a significant challenge. Thus, in the present study, to deliver p66shc siRNA into spinal cord effectively and to reduce the number of treatment with p66shc siRNA, we performed the intrathecal administration with poly(d,l-lactic-co-glycolic acid) (PLGA) nanoparticles (NPs) containing p66shc siRNA (p66shc siRNA-PLGA NPs), because PLGA-based drug products are designed to reduce dosing frequency and potential drug toxicity, and siRNA-loaded NPs for gene therapy in various neurological diseases have attracted a great deal of attention. In addition, to verify the role of p66shc, we also examined the effects of p66shc siRNA on H2O2-induced oxidative stress in HT22 neuronal cells. According to the reviewer’s comment, we modified the last paragraph of the Introduction section. Please refer to the text.
Comment 13: Except few, most of the references are old. Please cite current references.
Response: According to the reviewer’s comment, we checked the whole references again, and we changed the old references with current references. Please refer to the text.
Comment 14: Level of English should be improved further.
Response: We already checked our manuscript by a professional English editing service before submission. According to the reviewer’s comment, we checked our manuscript again.
Comment 15: Abstract and conclusion should be improved further.
Response: According to the reviewer’s comment, we improved it. Please refer to the text.
Comment 16: Plagiarism of manuscript is high, please revise it carefully especially experimental part.
Response: We are sorry about high plagiarism. However, as the reviewer can find in the pubmed, we performed the same experimental protocols, and we reported many articles about SNL-induced neuropathic pain and other experimental models with/without using PLGA nanoparticles. According to the reviewer comment, we checked our manuscript thoroughly, cited with our previous studies, and rewrote some sentences to avoid plagiarism, although some parts could not be different with our previous articles perfectly. Please comprehend this and refer to the text.
We deeply thank again for the prudent comments on our manuscript.
Very sincerely yours,
Corresponding Author:
Choong-Hyun Lee, DVM, PhD.
Department of Pharmacy, College of Pharmacy, Dankook University, Cheonan, 31116, Republic of Korea.
TEL: +82-41-550-1441. FAX: +82-41-559-7899.
E-mail: [email protected]

Reviewer 3 Report
Main Comments
Abstract, line 4. …..neuropathic pain in rats. The SNL……
Abstract, line 5. …..compared with the scrambled siRNA-treated SNL group. Cleaved….
Abstract, line 6. ….decreased in the L5 spinal cord…..
Abstract, line 9. ….replace the word ‘indicate’ by ‘suggest’
Abstract, line 11. ….processes following SNL in rats.
Introduction, para 3, line 4. ….nerve ligation (SNL) in rats.
Introduction, para 3, line 5. ….we used intrathecally administered…..
Section 2.2 entitled ‘Neuropathic pain model and behavioural test’
Para 1, line 4. How many 3-0 silk sutures were tightly ligated around L5?
Para 1, line 5. Add a sentence about the monitoring of rats during post-surgical recovery.
Para 2, line 1. Add name of supplier for the von Frey filaments. Were pain behavioural assessments performed by testers who were blinded to the surgery performed and the test treatment? If so, say so.
Para 2, line 5, last sentence. On what post-surgical day were rats tested to decide whether or not they had neuropathic pain? What was the success rate for inducing neuropathic pain?
Section 2.2 entitled ‘Preparation and intrathecal injection…..’
Para 1, line 2. ….Give some details here on the method. Where was PLGA purchased from and what was its molecular weight?
Para 1, line 5. …the acronym ‘TE’ has not been defined.
Para 1, line 5. …What grade of DCM was used and where was it purchased from?
Para 1, line 6. ….the acronym ‘PVA’ is not defined. What grade was used and where was it purchased from?
Para 1, line 9. ….What was the the siRNA loading and encapsulation efficiency of the NPs? How were these parameters measured? Give the method details.
Para 1, line 10. ….For how long were the NPs freeze-dried. What do you mean by the phrase ‘freeze-dried with 10 vials’?
Para 1, line 11. ….what percentage of isofurane? Was it delivered in oxygen?
Para 1, line 12. ….How was the concentration of 4mM siRNA verified? Give the dose administered in nmol.
Section 2.5 entitled ‘Tissue processing and immunohistochemistry’
Line 1. ….How were rats euthanised prior to perfusion fixing with 4% PFA?
Section 2.6 entitled ‘Cell cultures’
There is insufficient detail in this section for your readers to be able to follow what you did; please re-write. What is SHC1HSS? What is scRNA? The meaning of the final sentence is unclear. What did you do with the cells after incubation with H2O2 for 3h?
Section 3.1 entitled ‘Characterisation of p66shc siRNA PLGA NPs’
What were the encapsulation efficiencies and the siRNA and scrambled siRNA loading for the NPs?
How was the siRNA release assay of NPs done? The detail of this method needs to be added to the Methods section.
Add the ‘release’ data as an extra panel in Figure 1.
In panel C there are 2 SEM images. Are these for the p66shc siRNA and the scrambled siRNA or something else? Please label these images appropriately.
For Figure 1, define the acronym ‘d.nm)….is it diameter in nm?
Section 3.2 entitled ‘Effect of p66shc siRNA-PLGA NPs on the SNL-induced mechanical hypersensitivity’
Line 2. How many rats per treatment group?
Figure 2, panel B. The degree of pain relief is partial at best as the target mechanical threshold is approximately 12g. Why didn’t you administer a larger dose to show dose-response and produce more pronounced anti-allodynia?
Section 3.3 entitled ‘Changes in p66shc and markers for apoptosis, autophagy and mitophagy following SNL?
Para 1, line 3. ….In the L5 dorsal horn of the spinal…
Section 3.4 entitled ‘Activation of microglia in the ipsilateral spinal cord following SNL’
Para 1, line 2. ….L5 dorsal horn of the spinal….
Figure 3 is mis-labelled ‘Figure 4’ and vice versa.
Figure 3 legend. What about the scrambled siRNA-SNL group?
Section 3.5 entitled ‘Reduced protein levels of p-p66shc and markers of……’
Line 1. Need to mention control group that received the scrambled siRNA.
Figure 4 legend. Did the SNL group receive scrambled siRNA? This needs to be made clear.
Figure 5 legend. Were any cells incubated with the scrambled siRNA?
Line 5. ……HT22 cells under…..
Figure 6 legend. …..resulted in decreased mRNA levels….
All of the acronyms used in Fig 6 need to be defined in the legend or as a footnote to the figure.
Discussion section.
There is a tendency to over-interpret the results.
Para 2, line 4. Replace word ‘effectively’ by ‘partial’
Para 2, line 5. Replace word ‘indicated’ by ‘suggested’
Para 3, 3rd last line. ….it is likely that the partial analgesic effect…..
Para 4, 3rd last line. ….in cultured HT22 cells.
Conclusion paragraph, line 2. NPs partially attenuated….
Minor Comments
Section 2.2 entitled ‘Preparation and intrathecal injection…..’
Para 1, line 12. ….replace ‘intratracheally’ by ‘intrathecally’
Author Response
Manuscript ID: Polymers-762381
Title: p66shc siRNA-encapsulated PLGA nanoparticles ameliorate neuropathic pain following spinal nerve ligation
Dear Reviewer
I appreciate the reviewer’s prudent comments on our manuscript. The responses to the reviewer’ comments are summarized below:
Response to Reviewer’s Comment:
Comment 1: Abstract, line 4. …..neuropathic pain in rats. The SNL……; line 5. …..compared with the scrambled siRNA-treated SNL group. Cleaved….; line 6. ….decreased in the L5 spinal cord…..; line 9. ….replace the word ‘indicate’ by ‘suggest’; line 11. ….processes following SNL in rats.
Response: According to the reviewer’s comment, we corrected it. Please refer to the text.
Comment 2: Introduction, para 3, line 4. ….nerve ligation (SNL) in rats.; para 3, line 5. ….we used intrathecally administered…..
Response: We rewrote this part more clearly. Please refer to the text.
Comment 3: Section 2.2. Para 1, line 4. How many 3-0 silk sutures were tightly ligated around L5?
Response: In this study, we ligated tightly only L5 nerve 3 times with 3-0 silk thread. Please refer to the text.
Comment 4: Section 2.2. Para 1, line 5. Add a sentence about the monitoring of rats during post-surgical recovery.
Response: According to the reviewer’s comment, we described the the monitoring of rats during post-surgical recovery. Please refer to the text.
Comment 5: Section 2.2. Para 2, line 1. Add name of supplier for the von Frey filaments. Were pain behavioural assessments performed by testers who were blinded to the surgery performed and the test treatment? If so, say so.
Response: In this study, we assessed the SNL-induced pain behavior by two investigators who did not perform the surgery as well as the intrathecal injection. According to the reviewer’s comment, we described this and added the name of supplier for the von Frey filaments. Please refer to the text.
Comment 6: Section 2.2. Para 2, line 5, last sentence. On what post-surgical day were rats tested to decide whether or not they had neuropathic pain? What was the success rate for inducing neuropathic pain?
Response: In this study, we assessed and decided whether or not they had neuropathic pain on the 3 days after surgery, and we found that the success rate of neuropathic pain modeling was about 80 %. According to the reviewer’s comment, we described this. Please refer to the text.
Comment 7: Section 2.3. Para 1, line 2. Give some details here on the method. Where was PLGA purchased from and what was its molecular weight?
Response: In this study, we used PLGA (Purasorb, PDLG 5002A, Corbion Amsterdam) products for PLGA nanoaprticles synthesis. PURASORB PDLG 5002A is an acid terminated GMP grade copolymer of DL-lactide and Glycolide in a 50/50 molar ratio and with an inherent viscosity midpoint of 0.2 dl/g. Their molecular formula is [(C6H8O4)x(C4H4O4)y]n, their molecular weight is measured by GPC, and their specification is for information purpose only. If we also don’t know its exact molecular weight, we describe the specific catalog number, according to the reviewer’s comment. Please comprehend this and refer to the text.
Comment 8: Section 2.3. Para 1, line 5. the acronym ‘TE’ has not been defined.
Response: According to the reviewer’s comment, we corrected it. Please refer to the text.
Comment 9: Section 2.3. Para 1, line 5. What grade of DCM was used and where was it purchased from?
Response: According to the reviewer’s comment, we described this. Please refer to the text.
Comment 10: Section 2.3. Para 1, line 6. the acronym ‘PVA’ is not defined. What grade was used and where was it purchased from?
Response: In this study, we used Polyvinyl alcohol (98-99% hydrolyzed, high molecular weight, CAS: 9002-89-5, Alfa Aeasar, Fisher Scientific). According to the reviewer’s comment, we described this. Please refer to the text.
Comment 11: Section 2.3. Para 1, line 9. What was the siRNA loading and encapsulation efficiency of the NPs? How were these parameters measured? Give the method details.
Response: As described in the text, PLGA NPs carrying p66shc siRNA or scrambled siRNA were prepared. In addition, we also measured the encapsulation efficiency of p66shc siRNA-PLGA NPs, according to the method of the previous studies (Shin et al. Polymers (Basel) 2020, 12(2) pii: E443, doi: 10.3390/polym12020443; Peltonen et al. AAPS PharmSciTech 2004, 5: E16). The percentage of encapsulation efficiency was calculated as the ratio between the amount of siRNA released from the PLGA NPs and the mount of siRNA initially taken to prepare the PLGA NPs, and we found that encapsulation efficiency was 32.3 %. Please refer to the text.
Comment 12: Section 2.3. Para 1, line 10. For how long were the NPs freeze-dried. What do you mean by the phrase ‘freeze-dried with 10 vials’?
Response: We are sorry about the obscure description. According to the reviewer’s comment, we corrected it. Please refer to the text.
Comment 13: Section 2.3. Para 1, line 11. what percentage of isofurane? Was it delivered in oxygen?
Response: According to the reviewer’s comment, we described this. Please refer to the text.
Comment 14: Section 2.3. Para 1, line 12. How was the concentration of 4mM siRNA verified? Give the dose administered in nmol.
Response: First of all, we made a big mistake. The concentration of siRNA was 1.6 μM, not 4 mM. When preparation, the concentration of siRNA 20 μM is contained in 1 vial. Before intrathecal injection, the each 20 μM of siRNA-encapsulated PLGA NPs in 1 vial were prepared with 250 μL PBS. At this point in time, the concentration was 20 μM/250 μL (= 80 nM/μL). Because we administered 20 μL siRNA, it can be calculated as 1,600 nM/20 μL. Please comprehend our mistake and refer to the text.
Comment 15: Section 2.5. Line 1. How were rats euthanised prior to perfusion fixing with 4% PFA?
Response: In this study, we anesthetized by intraperitoneal administration with sodium pentobarbital (100 mg/kg) before perfusion. We described this. Please refer to the text.
Comment 16: Section 2.6. There is insufficient detail in this section for your readers to be able to follow what you did; please re-write. What is SHC1HSS? What is scRNA? The meaning of the final sentence is unclear. What did you do with the cells after incubation with H2O2 for 3h?
Response: We are sorry about the insufficient description. According to the reviewer’s comment, we rewrote this part. Please refer to the text.
Comment 17: Section 3.1. What were the encapsulation efficiencies and the siRNA and scrambled siRNA loading for the NPs?
Response: we think that this comment is similar with comment 11.
Comment 18: Section 3.1. How was the siRNA release assay of NPs done? The detail of this method needs to be added to the Methods section.
Response: According to the reviewer’s comment, we described the siRNA release assay. Please refer to the text.
Comment 19: Section 3.1. Add the ‘release’ data as an extra panel in Figure 1.
Response: According to the reviewer’s comment, we added the release data in the new Figure 1. Please refer to the new Figure 1.
Comment 20: Section 3.1. In panel C there are 2 SEM images. Are these for the p66shc siRNA and the scrambled siRNA or something else? Please label these images appropriately.
Response: According to the reviewer’s comment, we represented this. Please refer to the new Figure 1.
Comment 21: Section 3.1. For Figure 1, define the acronym ‘d.nm)….is it diameter in nm?
Response: According to the reviewer’s comment, we represented this. Please refer to the new Figure 1.
Comment 22: Section 3.2. Line 2. How many rats per treatment group?
Response: According to the reviewer’s comment, we represented this in the Methods section and in the Figure 2.
Comment 23: Figure 2, panel B. The degree of pain relief is partial at best as the target mechanical threshold is approximately 12g. Why didn’t you administer a larger dose to show dose-response and produce more pronounced anti-allodynia?
Response: In our previous study (Yin et al. Int J Mol Sci 2019, 20:4443), we observed that gabapentin, currently used as a neuropathic pain reliever, was most effective 2 hours after administration, but its effectiveness was confirmed within a 1 day, and that it recovered to a mechanical threshold level of about 6 g. As the reviewer knows, generally, the value of the pain assessment experiment is about 4 g. From the result of our present study, we could find that our gene therapy products were sufficiently effective. Please comprehend this.
Comment 24: Section 3.3. Para 1, line 3. ….In the L5 dorsal horn of the spinal…
Response: According to the reviewer’s comment, we corrected it more clearly as follows: In the ipsilateral dorsal horn of the L5 spinal…. Please refer to the text.
Comment 25: Section 3.4. Para 1, line 2. ….L5 dorsal horn of the spinal….
Response: According to the reviewer’s comment, we corrected it. Please refer to the text.
Comment 26: Figure 3 is mis-labelled ‘Figure 4’ and vice versa.
Response: We checked it again carefully.
Comment 27: Figure 3 legend. What about the scrambled siRNA-SNL group?
Response: As described in the section 3.2. of the Result section, SNL group means the scrambled siRNA-PLGA NP treatment group after SNL. Please refer to the Result section.
Comment 28: Section 3.5. Line 1. Need to mention control group that received the scrambled siRNA.
Response: As the reviewer mentioned, control group means the scrambled siRNA-treated group. According to the reviewer’s comment, we described this. Please refer to the text.
Comment 29: Figure 4 legend. Did the SNL group receive scrambled siRNA? This needs to be made clear.
Response: As responded in the comment 29, SNL group means the scrambled siRNA-PLGA NP treatment group after SNL, as described in the section 3.2. of the Result section. Please refer to the Result section.
Comment 30: Figure 5 legend. Were any cells incubated with the scrambled siRNA?
Response: We responded this in the above comment 28.
Comment 31: Figure 5 legend. Line 5. ……HT22 cells under…..
Response: According to the reviewer’s comment, we corrected it. Please refer to the text.
Comment 32: Figure 6 legend. …..resulted in decreased mRNA levels….
Response: According to the reviewer’s comment, we corrected it. Please refer to the text.
Comment 33: All of the acronyms used in Fig 6 need to be defined in the legend or as a footnote to the figure.
Response: According to the reviewer’s comment, we defined all of the acronyms in the legend of the Fig 6 as well as in the section 2.7. of the Materials and Methods section. Please refer to the text.
Comment 34: Discussion section. There is a tendency to over-interpret the results. Para 2, line 4. Replace word ‘effectively’ by ‘partial’; Para 2, line 5. Replace word ‘indicated’ by ‘suggested’; Para 3, 3rd last line. ….it is likely that the partial analgesic effect…..; Para 4, 3rd last line. ….in cultured HT22 cells.
Response: According to the reviewer’s comment, we corrected it. Please refer to the text.
Comment 35: Conclusion paragraph, line 2. NPs partially attenuated….
Response: According to the reviewer’s comment, we corrected it. Please refer to the text.
Comment 36: Section 2.2. Para 1, line 12. ….replace ‘intratracheally’ by ‘intrathecally’
Response: According to the reviewer’s comment, we corrected it. Please refer to the text.
We deeply thank again for the prudent comments on our manuscript.
Very sincerely yours,
Corresponding Author:
Choong-Hyun Lee, DVM, PhD.
Department of Pharmacy, College of Pharmacy, Dankook University, Cheonan, 31116, Republic of Korea.
TEL: +82-41-550-1441. FAX: +82-41-559-7899.
E-mail: [email protected]

Round 2
Reviewer 1 Report
I think that the manuscript has been significantly improved and now warrants publication in Polymers.Reviewer 3 Report
The authors have suitably addressed the Reviewer's comments.